# Efficacy and Safety of Continuous Infusion of Vancomycin in Children: A Systematic Review

**DOI:** 10.3390/antibiotics10080912

**Published:** 2021-07-26

**Authors:** Marta Alonso-Moreno, Marta Mejías-Trueba, Laura Herrera-Hidalgo, Walter Alfredo Goycochea-Valdivia, María Victoria Gil-Navarro

**Affiliations:** 1Unidad de Gestión Clínica de Farmacia, Hospital Universitario Virgen del Rocío, 41013 Seville, Spain; marta.alonso.moreno.sspa@juntadeandalucia.es (M.A.-M.); marta.mejias.sspa@juntadeandalucia.es (M.M.-T.); 2Unidad de Gestión Clínica de Farmacia, Instituto de Biomedicina de Sevilla (IBiS), Hospital Universitario Virgen del Rocío, 41013 Seville, Spain; mariav.gil.sspa@juntadeandalucia.es; 3Unidad de Pediatría de Enfermedades Infecciosas, Reumatología e Inmunología, Instituto de Biomedicina de Sevilla (IBiS), Hospital Universitario Virgen del Rocío, 41013 Seville, Spain; alfgova@hotmail.com

**Keywords:** vancomycin, continuous, infusion, pediatrics, children

## Abstract

Vancomycin is used to treat a wide variety of infections within the pediatric population. In adults, continuous infusion of vancomycin (CIV) has been evaluated as an alternative to intermittent infusion of vancomycin (IIV) with potential advantages. In children, the use of CIV is increasing; however, data is currently limited. The objective is to provide efficacy and safety evidence for CIV within this population. The review was carried out following PRISMA guidelines. A bibliographic search was performed for studies on PubMed and EMBASE. Clinical trials and observational studies that reported clinical efficacy and/or target attainment of CIV in pediatrics were included. Articles were reviewed to assess their design and target population, characteristics of vancomycin treatment and the main findings in terms of safety and efficacy. A total of 359 articles were identified, of which seven met the inclusion criteria. All of them evaluated the target attainment, six assessed safety but only three assessed clinical efficacy. The best administration method for this antibiotic within the pediatric population is still unknown due to limited evidence. However, studies conducted thus far suggest pharmacokinetic advantages for CIV. Further investigation is required, in particular for studies comparing IIV with CIV for clinical efficacy and toxicity outcomes.

## 1. Introduction

Vancomycin is a glycopeptide antibiotic used to treat a wide variety of systemic Gram-positive infections, including methicillin resistant *Staphylococcus aureus* (MRSA) and methicillin resistant coagulase-negative *Staphylococcus* (MRCNS) in adult and pediatric populations [1,2]. Vancomycin exhibits time-dependent bactericidal activity, meaning that the time in which the concentration of the drug in the body is above the minimum inhibitory concentration (MIC) affects antimicrobial efficacy [2,3].

In adults, the vancomycin area under the plasma concentration-time curve (AUC) to MIC ratio (AUC/MIC) > 400 has long been the best predictor of clinical and bacteriological efficacy for patients with severe infections caused by MRSA [4]. Recently, a revised consensus guideline developed by different scientific associations has been published, recommending a target of an AUC/MIC ratio of 400 to 600 (assuming a MIC of 1 mg/L) for empiric dosing in both adult and pediatric patients to maximize clinical efficacy and minimize nephrotoxicity [5]. However, there is a lack of evidence for this parameter in children due to the complexity of vancomycin clearance in the various pediatric age groups, and the differences in tissue site-of-infection drug exposure as a consequence of higher pharmacokinetic variability [5,6]. Due to the impracticalities of calculating the AUC, target trough concentrations of 15 to 20 mg/L are used as a surrogate marker in adults with normal renal function when MIC is ≤1 mg/mL [7,8]. For the pediatric population, there is more controversy in establishing a target trough concentration. The majority of studies suggest a trough concentration between 6–11 mg/L to achieve AUC/MIC > 400, however no consensus has been reached [3,9,10].

In adults, continuous infusion of vancomycin (CIV) has been evaluated as an alternative to intermittent infusion of vancomycin (IIV) with potential advantages including: earlier concentration target attainment, less variability in serum concentrations, ease of drug level monitoring, and lower risk of nephrotoxicity [5,11,12]. When compared to adults, achieving therapeutic serum vancomycin concentrations (SVCs) with IIV in children requires higher doses and shorter intervals given their increased renal clearance [9,13]. However, higher doses have also been associated with increased nephrotoxicity in pediatrics [6].

Consequently, the use of CIV in children is increasing through a number of heterogenous practices, despite limited efficacy and safety data in this population [13,14,15]. 

This systematic review aims to provide efficacy and safety evidence for CIV within the pediatric population

## 2. Results

### 2.1. Bibliographic Search

A total of 359 articles were obtained from the different databases (72 from PubMed and 287 from EMBASE). After eliminating 55 duplicates using Mendeley checking, a total of 304 articles were left.

After title and abstract screening, a further 277 studies were deemed ineligible because they did not meet the inclusion criteria according to the PICOS question, mainly because it did not include a pediatric population or evaluate the pharmacokinetics of other antibiotics. The 27 potentially relevant studies were reviewed in full text, of which 20 were excluded before data extraction. Seven met the inclusion criteria and were therefore included in this systematic review (Figure 1).

### 2.2. Quality of the Included Studies

The methodological quality of the studies included in this review was variable. One study was evaluated as good quality, two studies were assessed as having some concerns or fair quality and four studies were reported as poor quality or with serious risk of bias. The detailed results for the risk of bias assessment are summarized in Table 1.

### 2.3. Characteristics of the Included Studies

Of the seven studies included, only one [16] was a randomized controlled trial. Of the remaining studies, two were retrospective studies [13,15], one was a prospective study [17], and three were case series studies [18,19,20]. The included studies were published between 2012 and 2019. Table 2, Table 3 and Table 4 present the variables and results of the articles included in this review.

### 2.4. Characteristics of Vancomycin Treatment

To calculate initial total daily dose, all studies considered patient bodyweight. Two studies also included age [13,17] and the study by Berthaud et al. [16] took into account three covariates: bodyweight, age, and serum creatinine. For subsequent doses, SVCs were considered in all studies.

In four studies, IIV was administered before conversion to CIV [13,18,19,20]. The total daily dose for patients decreased when switching to CIV in all cases. Within the remaining studies [15,16,17], patients were given a CIV dosage regimen directly.

Only two studies [15,16] administered a loading dose before starting CIV therapy. The dose used was 12–16 mg/kg and neither of these studies used IIV beforehand.

In all studies except one [16], the majority of patients (54–100%) used vancomycin as targeted antimicrobial therapy, with MRSA and bloodstream infection being the most frequently isolated microorganism and site of infection, respectively.

### 2.5. Target Attainment

All identified studies except one [16], evaluated pharmacokinetic efficacy as serum vancomycin target concentration (SVTC) attainment. Berthaud et al. [16] evaluated pharmacokinetic efficacy with AUC/MIC ratio. SVTC was a secondary endpoint in this study. The therapeutic SVCs ranges that were predefined in the different studies were variable (10–40 mg/L) (Table 4). Four studies [13,18,19] predefined the therapeutic SVCs range between 15–20 mg/L, however the percentage of patients with SVTC were quite variable: 0% [18], 27% [19], 58.5% [17] and 59% [13]. In the three case series studies [18,19,20] that used IIV before CIV, suboptimal SVCs were obtained with conventional IIV in the majority of patients, therefore in all cases, pharmacokinetic results obtained were better with CIV compared to IIV. One case in particular, managed to achieve 100% SVTC [20]. In the Fung et al. case series study [18], although theoretically no patients achieved SVTC, they all obtained SVCs closer to target when switching from IIV to CIV, which correlated with improved clinical efficacy.

Most of the studies [13,15,16,17,18,19] measured SVCs within the first 24–48 h. Zylbersztajn et al. showed no data on SCVs timing [20] while Hoegy et al. [17] measured it between 48 h and 96 h after therapy initiation.

Only three studies [15,16,18] evaluated the percentage of patients who achieved an AUC/CMI > 400. One of them [16] established AUC/CMI ratio as a primary pharmacological target for vancomycin with a target range between 400 and 800 mg·h/L to maximize clinical efficacy and minimize nephrotoxicity.

### 2.6. Clinical Efficacy

Three articles [16,18,20] included in this review evaluated the clinical efficacy of CIV vancomycin treatment. However, none of them were designed for this purpose. 

The first one [16] analyzed clinical efficacy only for patients with a final diagnosis of infection treatable by vancomycin and who had received vancomycin for at least seven days. No statistically significant differences were found between patients treated with CIV with and without an early Bayesian dose adjustment regarding sustainable apyrexy, C-reactive protein reduction and bacteraemia duration. Fung et al. [18] evaluated three patients with cystic fibrosis and concluded that there had been a clinical improvement in all of them. Finally, Zylbersztajn et al. [20] reported six cases of patients treated with CIV. They observed clinical improvement in all six patients, with four of these also showing microbiological cure.

### 2.7. Safety

All articles, except one [17] evaluated nephrotoxicity. Two articles [16,19] defined nephrotoxicity as an increase in serum creatinine (SCr) ≥ 0.5 mg/dL or a 50% increase from the baseline SCr. One study [13] evaluated effects on renal function using the RIFLE (risk, injury, failure, loss of kidney function, and end stage kidney disease) classification [21]. Two studies [15,20] exclusively evaluated SCr levels and Fung et al. [18] evaluated variation in SCr levels (on admission and prior to discharge), blood urea nitrogen and urine output.

Cases of nephrotoxicity were observed only in two out of six studies [15,16], giving results of 11% and 12% respectively.

Only three studies evaluated immediate adverse events. McKamy et al. [19] evaluated phlebitis, peripheral line loss, and infusion reactions. The frequencies of these adverse reactions to CIV therapy were low and not specified. Berthaud et al. [16] evidenced four cases of red man syndrome and three deaths, however, none of the deaths were attributed to infection nor to iatrogenic events. Finally, Hurst et al. [13] did not note any patients who had any infusion reactions while on CIV therapy.

## 3. Discussion

This systematic review shows that there remains a lack of evidence for CIV use in pediatrics. Distinct to adults, vancomycin therapeutic drug monitoring adjustment presents a higher level of complexity in children, characterized by unique pharmacokinetic parameters due to important physiological changes because of their rapid development with an increased renal clearance [22].

Understandably, these unique characteristics may support CIV use in certain clinical conditions, since this seems to facilitate the achievement and maintenance of therapeutic SVCs avoiding high nephrotoxic doses as shown in this review. However, the limited data available in this vulnerable population has not allowed its routine use to be generalized in clinical practice and to suggest an optimal dosing regimen for pediatric CIV.

Most of the identified literature recommends CIV use in patients who were unable to achieve therapeutic SVCs or desired clinical outcome with IIV [13,18,19,20]. Yet, there is no prediction model to identify such patients in advance, which can lead to the loss of important days of therapy under suboptimal IIV regimens. The delay in optimizing therapeutic SCVs could result in treatment failure that may influence morbidity and mortality outcomes in those suffering severe infections. Furthermore, alternative antimicrobial therapy options could be wrongly selected if vancomycin treatment failure is suspected, when rather than spectrum or activity, attainment of SVTC is the main problem. This could develop a negative ecological impact and should be actively assessed by antimicrobial stewardship programs. For example, in the case series study of McKamy et al. [19], all patients except for one were converted from IIV to CIV therapy within four to seven days of the initiation of treatment with vancomycin after two consecutive suboptimal therapeutic SVCs. Also, in a retrospective observational study [13], therapeutic SVCs were not achieved even with doses higher than 80 mg/kg/day using IIV; however, when therapy was converted to CIV, 65% of patients achieved SVTC (78% in the goal SVC 10 to 15 mg/L group and 59% in the goal SVC 15 to 20 mg/L group). The initial dosing used for CIV therapy was 56% to 60% less than the final daily dosing of IIV therapy in both goal SVC groups (10–15 mg/L or 15–20 mg/L), but statistically significant higher initial CIV dosing was used in the younger age groups (*p* = 0.023 and 0.002 for the 10–15 and 15–20 mg/L goal SVC groups, respectively) [13]. This can be explained as age is a predictor of vancomycin clearance within the pediatric population, and younger patients have lower trough concentrations than older patients receiving the same dose [23]. Hoegy D et al. suggest that it is necessary to prescribe certain dosage regimens for each age group, being higher for young age groups [17].

Loading dose administration prior to CIV initiation can also improve the time to achieve SVTC. Two studies [15,16] used loading doses of 12 to 16 mg/kg to rapidly obtain desired steady-state SVTC; however, success in achieving the target ranges was dependent on the use of adequate initial CIV doses. Genuini et al. [15], observed that less than 50% of children achieved SVTC using the recommended dosing regimens, which authors described as an inappropriate result. Consequently, Genuini et al. [15], proposed the use of a pharmacokinetic model with a covariate-adjusted starting dose and Bayesian estimation to achieve the pharmacokinetic target in future studies. Subsequently, one year later, a randomized controlled trial [16] was conducted using a population-based PK (POPPK) model published by Le et al. in 2013 [24] that included age, bodyweight, and serum creatinine as covariates. This study achieved the vancomycin target in more than 50% of children (85% from the Bayesian group and 57% from the control group), showing that covariates could affect PK parameters that may be important to intersubject variability, and its use would be advisable to individualize CIV dose a priori.

The optimal approach for vancomycin therapeutic monitoring is not well established, but recent guidelines support AUC-based monitoring [5]. However, AUC-based monitoring is complicated with conventional IIV, so trough SVCs are used in clinical practice as a surrogate marker for the optimal vancomycin AUC/MIC > 400 if the MIC is ≤1 mg/L in patients with normal renal function. Trough SVCs in IIV cannot be extrapolated to CIV. Thus, for a target of AUC/MIC > 400–600, a SVCs of 10–15 mg/L is accepted in mild–moderate infections and 15–20 mg/L in severe infections or those with difficult access. However, in CIV a concentration of 20–25 mg/L is recommended for an expected AUC/MIC of 480–600 (assuming a MIC of 1 mg/L) [5].

Within the studies analyzed in this review, there was variability in defining the values for target attainment with CIV, with all studies reporting a wide range interval (10–40 mg/L). One study [16] used AUC/MIC > 400 as a primary pharmacological target for vancomycin, instead of an inaccurate approximation using SVCs that poorly correlate to exposure. Based on current available data, the proposal for AUC-guided monitoring in pediatrics aligns with the approach for adults, including the application of Bayesian estimation for one trough concentration or first-order PK equations with two concentrations [5]. Similarly, unlike intermittent administration, calculating AUC with CIV requires only one steady-state SVC, which can be scheduled along with other scheduled laboratory draws at any time [2].

In this review, most of the studies conducted therapeutic monitoring within 24 to 48 h of CIV therapy. This is recommended by clinical practice guidelines, with delays in therapeutic monitoring made depending on the severity of infection and clinical judgment [5]. Besides, a randomized controlled trial [16] has showed that early Bayesian dose adjustment at 6 h significantly and safely increased pharmacological target attainment in children at the 24 h of treatment with CIV, which could improve clinical and bacteriological outcomes for MRSA infections in this particular population [16].

Clinical and microbiologic efficacy were not evaluated in most studies. Just two small clinical case series without control group [18,20] showed a clinical improvement and negative blood cultures in the majority of patients. A randomized controlled trial [16] compared clinical outcomes between Bayesian and control groups, both with CIV, and there were no differences between groups. However, this clinical trial was not designed to evaluate this. In adults, there is no evidence to indicate that CIV is clinically superior to IIV. However, studies provide evidence supporting that the complexities of dosing and monitoring IIV can be attenuated by CIV therapy [25]. CIV is associated with lower variabilities in the serum concentration and favourable SVTC attainment, which translates into an improved vancomycin exposure, and is currently a better predictor of clinical efficacy [12].

Regarding CIV safety, nephrotoxicity was reported only in a small percentage of patients and was reversible in all cases [15,16]. Moreover, patients with vancomycin-attributable nephrotoxicity had vancomycin concentrations within the therapeutic range. This is in accordance with the hypothesis that elevated SVC is not the only predictive factor for vancomycin-induced renal injury, since it is known that intensive care unit admission, hypovolemia, concomitant administration of other nephrotoxic medications, such as aminoglycosides and diuretics, also contribute to the development of nephrotoxicity [6,15]. Although there is no evidence in children, several meta-analyses in adults have demonstrated that patients treated with CIV had a significantly lower incidence of nephrotoxicity compared with patients receiving IIV [11,26]. This can be explained as CIV minimizes vancomycin serum peak and maximizes trough concentrations, eliminating peak—trough variations of IIV and maintaining intermediate SVC once a steady-state is achieved. In these studies, CIV appeared to achieve a safer serum concentration profile when IIV and CIV dosing regimens were adjusted to achieve the same AUC [11,25,27]. The frequencies of immediate adverse reactions to CIV therapy, such as red man syndrome and phlebitis, were evaluated in three studies. One singular study [16] described the prevalence of red man syndrome as less than 5%, which suggests that it might be an alternative option for patients who experience infusion-related reactions with IIV dosing.

This systematic review of existing literature summarizes all the evidence published so far on CIV in children, analyzing important data and highlighting unresolved aspects that must be further studied to improve this valuable tool in pediatrics. The lack of neonatal data is a key limitation of this review; however, the neonatal population deserves its own analysis given high pharmacokinetics variability that is different from older children. Other limitations include high heterogeneity of the selected articles in terms of study design and quality that do not allow for pooled data analysis, and the lack of studies assessing clinical efficacy and adverse events other than nephrotoxicity.

## 4. Materials and Methods

A systematic review was carried out in accordance with the PRISMA guidelines (preferred reporting items for systematic reviews and meta-analysis) [28].

### 4.1. Selection Criteria

The inclusion criteria, defined according to the PICOS question (population, intervention, comparison, outcomes and study design), were the following:Population: pediatric patients (age range: ≥1 month and ≤19 years) receiving treatment with vancomycin.Intervention: continuous infusion of vancomycin.Comparison: with comparator or without comparator.Outcomes: clinical efficacy or SVCs target attainment, the latter defined as reaching target serum concentrations.Study Design: clinical trials and observational studies.

Studies with the following criteria were excluded: adults (age > 19 years) or the neonatal population (age < 1 month), data obtained from patients who received any form of extracorporeal circulation (renal replacement therapy, plasmapheresis, extracorporeal membrane oxygenation), studies with language other than English or Spanish, and studies with a case report design.

### 4.2. Data Sources

A literature search from inception until November 2020 of two electronic databases (MEDLINE, through the PubMed interface, and EMBASE) was performed. We used a combination of keywords associated with the concepts presented in Figure 2. The search strategy was amended according to the functionality of each of the databases.

### 4.3. Study Selection

Firstly, duplicate articles were eliminated. Thereafter, two reviewers (reviewer 1 and reviewer 2) independently selected the articles according to the inclusion criteria, based on the information obtained from the title and abstract. Relevant articles or those with insufficient information within the title and abstract were full text assessed.

In case of uncertainty, a third reviewer (reviewer 3) resolved any disagreement. A critical reading of the complete selected articles was then carried out.

### 4.4. Quality Assessment

To evaluate the quality of the studies selected for inclusion, we used three tools according to the study design. A Cochrane risk-of-bias tool for randomized trials (ROB.2) was used for randomized controlled trials [29] and the risk of bias in non-randomized studies of interventions tool (ROBINS-I) for non-randomized studies [30]. For case series studies, quality was assessed using a standardized study quality assessment tool (SQAT) designed by the National Heart, Lung, and Blood Institute, which forms part of the National Institute of Health in the United States [31]. When ROB.2 was applied, risk of bias was classified into the “Low”, “High” or “Some concerns” categories. When using the ROBINS-I tool, the overall risk of bias of the paper was categorized into “Low”, “Moderate”, “Serious” or “Critical”. Finally, when using the NIH quality assessment tool, the reported risk of bias was summarized as “Good”, “Fair” or “Poor”. Two independent reviewers (reviewer 1 and reviewer 2) conducted the quality assessment, and any disagreements on quality ratings between reviewers were discussed and then a consensus was reached.

### 4.5. Data Extraction

Reviewer 1 independently extracted standardized data and reviewer 3 examined all extraction sheets to ensure their accuracy and to minimize bias. A descriptive analysis of the main characteristics of the included studies was carried out and presented in tables.

The design and target population and purpose/objective of each of the identified studies were compiled in Table 2, including: type of study, comparator, study population (characteristics and number of patients, age, and sex) and main objective. Table 3 summarizes the information regarding characteristics of vancomycin treatment used in each article (therapy duration, microorganism treated, site of infection, type of treatment (empirical vs targeted) and total daily dose). Table 4 summarizes the main findings in terms of safety and efficacy. Safety is related to the presence of adverse events and efficacy is understood as clinical or microbiological cure or the achievement of therapeutic SVCs.

Given the heterogeneity of the study designs, interventions, and outcome measures, it was not feasible to pool the results in a meta-analysis. Alternatively, we performed a narrative synthesis of evidence following the Cochrane Consumers and Communication Review Group’s guidelines.

## 5. Conclusions

The use of CIV in children was reviewed. The best administration method for this antibiotic within the pediatric population in terms of efficacy and safety suggest pharmacokinetic advantages for CIV. Yet, several aspects remain to be defined, including optimal SVTC, best dose regimen, and ideal therapeutic drug monitoring and adjustment models with appropriate covariate selection.

Further investigation is required, preferably randomized clinical trials comparing IIV with CIV for clinical efficacy and toxicity outcomes, as well as timing to SVTC attainment and maintenance of AUC values. Correlation between pharmacokinetic measures and clinical/microbiological outcomes should be studied for both administration methods.

## Figures and Tables

**Figure 1 antibiotics-10-00912-f001:**
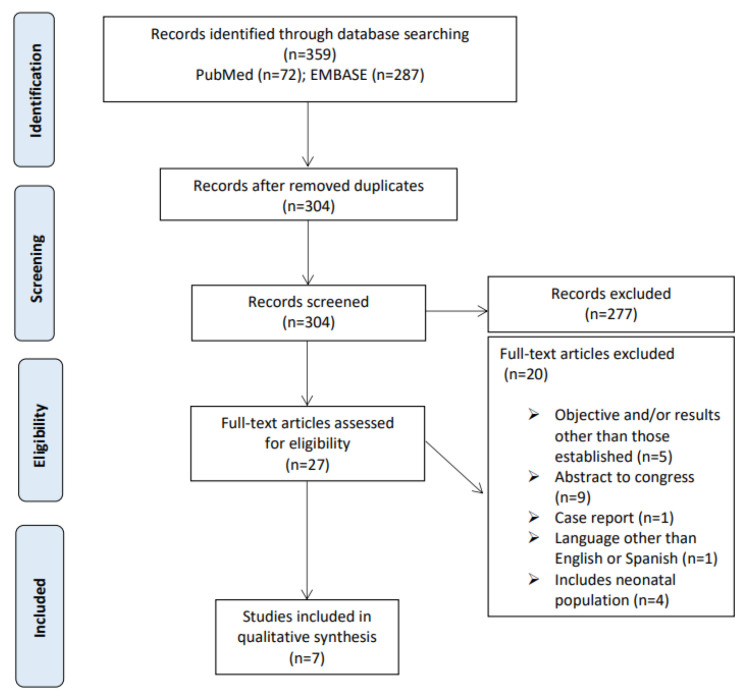
Study selection flowchart.

**Figure 2 antibiotics-10-00912-f002:**
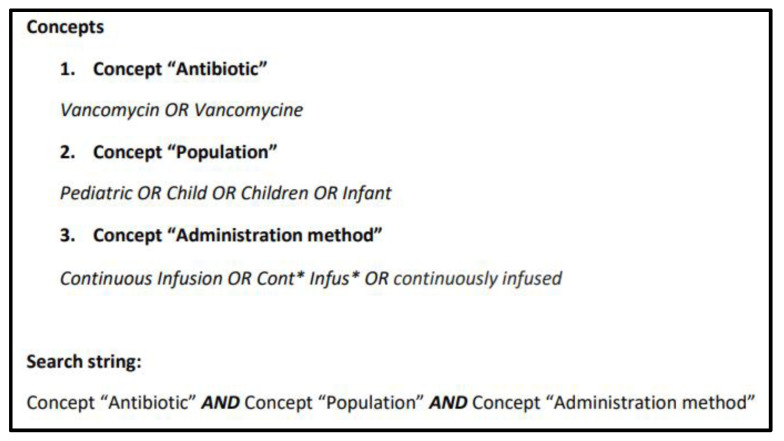
Search strategy.

**Table 1 antibiotics-10-00912-t001:** Quality assessment of the studies.

Risk of Bias Due To											
Study	Tool	Confounding	Selection of Participants	Classification of Interventions	Deviations from Intended Interventions	Missing Data	Measurement of Outcomes		Selection of the Reported Result		Overall Bias
Genuini M 2018 [15]	Robins	S	L	L	S	M	M		M		S
Hoegy D 2018 [17]		S	L	L	M	L	M		S		S
Hurst 2019 [13]		S	L	L	S	S	M		S		S
**Study**	**Tool**	**Randomization Process**		**Deviations from Intended Interventions**		**Missing Data**	**Measurement of Outcomes**		**Selection of the Reported Result**		**Overall Bias**
Berthaud R 2019 [16]	ROB-2	SC		SC		SC	L		SC		SC
**Study**	**Tool**	**Study Question**	**Population**	**Consecutive**	**Cases Comparable**	**Intervention**	**Measurement of Outcomes**	**Length of Follow up**	**Statistical Methods**	**Results**	**Overall**
Fung L 2012 [18]	SQAT	Y	Y	CD	N	Y	Y	Y	NA	Y	FAIR
Zylbersztajn BL 2013 [20]		N	Y	N	N	Y	N	Y	NA	N	POOR
McKamy S 2012 [19]		Y	Y	Y	N	Y	Y	Y	NA	Y	GOOD

Legend: L = Low/M = Moderate/S = Serious/SC = Some concerns/Y = YES/N = NO/NA = Not applicable/CD = Cannot determine. Colors: green = low risk; Yellow = moderate or some concerns risk; Orange = serious risk.

**Table 2 antibiotics-10-00912-t002:** Design and study population.

Reference	Type of Study	Comparator		Study Population		Main Objective
			Characteristics and Number of Patients	Age	Sex	
Berthaud R 2019 [16]	single-center randomized controlled trial	yes (control group without early Bayesian dose adjustment)	99 patients: 49 from Bayesian group 50 from control group	3 months-old to 17 years-old	48% (*n* = 48) males 52% (*n* = 51) females	to assess if an early Bayesian dose adjustment of vancomycin would increase the rate of target attainment in the first 24 h of treatment
Fung L 2012 [18]	case series study	no	3 patients with cystic fibrosis	3 years-old, 15 years-old and 17 years-old	100% (*n* = 3) females	to achieve therapeutic serum concentrations with continuous infusions of vancomycin
Genuini M 2018 [15]	Retrospective observational single-center study	no	28 critical ill patients	1 month-old to 17 years-old	21.42% (*n* = 6) males 78.58% (*n* = 22) females	Describe and assess a continuous infusion dosing scheme of vancomycin therapy in critically ill children
Hoegy D 2018 [17]	prospective study	no	94 patients hospitalized in hematology-oncology ward	4.3 months-old to 17.9 years-old	50% (*n* = 47) males 50% (*n* = 47) females	to prospectively validate an age-based dosing regimen for continuous IV vancomycin
Hurst AL 2019 [13]	Retrospective observational single-center study	no	240 patients (215 patients with CIV)	1 month-old to 18 years-old	60% (*n* = 144) males 40% (*n* = 100) females	to determine the total daily dose of CIV required to attain therapeutic serum vancomycin concentrations (SVCs) in pediatric patients according to age
McKamy S 2012 [19]	case series study	no	15 patients with pneumonia (*n* = 10) or osteomyelitis (*n* = 5)	6 months-old to 19 years-old (5.8 ± 6.1 years)	86.66% (*n* = 13) males 13.34% (*n* = 2) females	to assess adverse effects, the achievement of target plateau SVCs at steady state and the adequacy of the empirical dosing strategy.
Zylbersztajn BL 2013 [20]	casa series study	no	6 critical ill patients	2 months-old–7 years-old	66.66% (*n* = 4) males 33.33% (*n* = 2) females	to assess security and efficacy of CIV

Legend: CIV = continuous infusion vancomycin/SVCs = serum vancomycin concentrations.

**Table 3 antibiotics-10-00912-t003:** Characteristics of vancomycin treatment used.

Reference	Therapy Duration (Days)	Microorganism Isolated	Site of Infection	Empirical/Targeted Antimicrobial Therapy	Total Daily Dose on IIV before CIV	Final Total Daily Dose on CIV
Berthaud R 2019 [16]	At least 7	*Staphylococcus epidermidis* (*n* = 17), *Staphylococcus haemolyticus* (*n* = 3), *Staphylococcus hominis* (*n* = 3), *Staphylococcus capitis* (*n* = 1), non-typeable CNS (*n* = 1), methicillin-sensitive *Staphylococcus aureus* (*n* = 3), *Streptococcus mitis* (*n* = 2), group A streptococcus (*n* = 1), and *Enterococcus faecalis* (*n* = 1)	blood	34% (*n* = 28) targeted 72% (*n* = 71) empirical	NA	loading dose 14.9 (14.7–15) + 48 (44.6–59.2) mg/kg/day
Fung L 2012 [18]	14–21	MRSA	sputum	100% targeted	60–76 mg/kg/day	30–50 mg/kg/day
Genuini M 2018 [15]	4 (1–18)	CoNS (*n* = 11), *Enterococcus faecalis* (*n* = 1), *Brevibacterium casei* (*n* = 1), *Staphylococcus aureus* (*n* = 1), and *Enterococcus avium* (*n* = 1).	blood	54% (*n* = 15) targeted 46% (*n* = 13) empirical	NA	loading dose 14.8 (12–16) mg/kg + 44 (35–61) mg/kg/day
Hoegy D 2018 [17]	no data	no data	no data	no data	NA	<2 years-old: 56.5 ±13.5 mg/kg/day 2–6 years-old: 51.9 ± 10.6 mg/kg/day 6–12 years-old: 46.6 ± 10.8 mg/kg/day >12 years-old: 40.7 ± 11.8 mg/kg/day
Hurst AL 2019 [13]	no data	*Streptococcus species* (*n* = 68), CoNs (*n* = 64), and MRSA (*n* = 31) were the most frequently isolated	blood, urine, cerebral spinal fluid, aspirate, and/or wound swab	65% (*n* = 156) targeted 35% (*n* = 84) empirical	10–15 mg/L: <2 years-old: 79.5 ± 9.6 mg/kg/day 2–8 years-old: 79.1 ± 8.5 mg/kg/day >8 years-old: 72.5 ± 12.6 mg/kg/day 15–20 mg/L: <2 years-old: 77.9 ± 12.4 mg/kg/day 2–8 years-old: 78.7 ± 10.9 mg/kg/day >8 years-old: 72.9 ± 13.8 mg/kg/day	10–15 mg/L: <2 years-old: 48.4 ± 4.6 mg/kg/day 2–8 years-old: 45.6 ± 5.5 mg/kg/day >8 years-old: 39.4 ± 7.3 mg/kg/day 15–20 mg/L: <2 years-old: 47.7 ± 5.4 mg/kg/day 2–8 years-old: 46.8 ± 5.4 mg/kg/day >8 years-old: 43.6 ± 5.4 mg/kg/day
McKamy S 2012 [19]	15.3 ± 23.1	MRSA (*n* = 9) and *Streptococcus pneumoniae* (*n* = 3)	blood, urine	100% targeted	68.4 ± 5.8 mg/kg/day	44.5 ± 12.6 mg/kg/day
Zylbersztajn BL 2013 [20]	9–18	MRSA (*n* = 6)	blood	100% targeted	40–80 mg/ kg/day	50–60 mg/kg/day

Legend: CIV = continuous infusion vancomycin/NA = Non applicable/MRSA = Methicillin-resistant Staphylococcus aureus/CoNs = Coagulase-Negative staphylococci.

**Table 4 antibiotics-10-00912-t004:** Main findings of each of the identified studies.

Reference			Target Attaintment			Clinical Data	Safety Data
	SVTC (mg/L)	Measured SVC (Hours)	Therapeutic Concentrations in Range	Infra/Supratherapeutic SVC	AUC/MIC (mg·h/L)		
Berthaud R 2019 [16]	20–40	within the first 24 h	68% (*n* = 27) from the Bayesian group 38% (*n* = 16) from the control group	no data	AUC/MIC= 400–800: 85% (*n* = 34) from Bayesian group 57% (*n* = 27) from control group	sustainable apyrexia, C-reactive protein evolution and duration of bacteremia was not statistically different between groups (*n* = 43)	nephrotoxicity occurred in 12% (*n* = 10). Red man syndrome occurred in 4% (*n* = 4).
Fung L 2012 [18]	15–20	24 h after the initiation of therapy	0% (*n* = 0)	<15 mg/L: 33.33% (*n* = 1) >20 mg/L: 66.66% (*n* = 2)	AUC/MIC > 400: 66.66% (*n* = 2) AUC/MIC < 400: 33.33% (*n* = 1)	clinical improvement (*n* = 3)	no signs of nephrotoxicity
Genuini M 2018 [15]	15–30	within the first 48 h	first measured SVC: 43% (*n* = 12) second measured SVC 45% (*n* = 9)	no data	AUC/MIC > 400: 25% (*n* = 7)	no data	nephrotoxicity occurred in 11% (*n* = 3)
Hoegy D 2018 [17]	14–21	between 48 h and 96 h after the initiation of therapy	<2 years-old: 61.5% (*n* = 8) 2–6 years-old: 53.8% (*n* = 21) 6–12 years-old: 56.3% (*n* = 9) >12 years-old:65.4% (*n* = 17)	<14 mg/mL <2 years-old: 38.5% (*n* = 5) 2–6 years-old: 43.6% (*n* = 17) 6–12 years-old: 25.0% (*n* = 4) >12 years-old: 19.2% (*n* = 5) >21 mg/mL: < 2 years-old: 0% (*n* = 0) 2–6 years-old: 2.6% (*n* = 1) 6–12 years-old: 18.7% (*n* = 3) >12 years-old: 15.4% (*n* = 4)	no data	no data	no data
Hurst AL 2019 [13]	10–15 15–20	≥23 h after the initiation of therapy	10–15 mg/L: <2 years-old: 82% (*n* = 14) 2–8 years-old: 82% (*n* = 31) > 8 years-old: 67% (*n* = 14) 15–20 mg/L <2 years-old: 81% (*n* = 19) 2–8 years-old: 41% (*n* = 23) >years-old: 76% (*n* = 54)	no data	no data	no data	no signs of nephrotoxicity
McKamy S 2012 [19]	15–20	within the first 24–48 h	27% (*n* = 4)	<15 mg/L: 20% (*n* = 3) >20 mg/L: 53% (*n* = 8)	no data	no data	no signs of nephrotoxicity
Zylbersztajn BL 2013 [20]	10–25	no data	100% (*n* = 6)	NA	no data	clinical improvement (*n* = 6) and negative blood cultures (*n* = 4)	no signs of nephrotoxicity

Legend: SVC = serum vancomycin concentration/SVTC = serum vancomycin target concentration/NA = Non applicable/CIV = continuous infusion vancomycin/AUC/MIC = area under the plasma concentration-time curve/minimum inhibitory concentration.

## Data Availability

Data is contained within the article.

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
