# Peer review of "Efficacy and Safety of Continuous Infusion of Vancomycin in Children: A Systematic Review"

_antibiotics, 2021, doi:10.3390/antibiotics10080912_

Round 1

Reviewer 1 Report

General comments:

The authors made a review of existing data regarding efficacy and safety of continuous infusion of vancomycin in children. This work is useful as vancomycin is a widely used antibiotic with a narrow therapeutic target.

I thank the authors for this well written article.

The verbs tenses and citation format should be homogenized.

Detailed comments:

Line 40: “patients with severe infections caused by MRSA [4].” This reference seems to be wrong. The authors should recheck this.

Line 51: “The majority of studies suggest a trough concentration lower than 15 mg/L to achieve AUC/MIC>400”

The authors should give a defined interval instead of an opened one.

Line 70: “A further 277 articles were excluded based on their title and abstract”

The authors are asked to give details on the reasons for the exclusion of these articles.

Line 82-84: “One study was evaluated as good quality, two studies were assessed as having some concerns or fair quality and four studies were reported as poor quality or with serious risk of bias”

A quick look on these articles raised questions on the authors’ evaluation. Some points seems questionable, especially regarding the evaluation of the article by berthaud et al. in which some items classified with “some concerns” may be judged with a low risk of bias. A blinded reevaluation of each articles for risks of bias may be beneficial.

Line 163-164: “All identified studies evaluated pharmacokinetic efficacy as serum vancomycin target concentration (SVTC) attainment.”

This statement is inconsistent with line 269-271 : “One study [20] used AUC/MIC >400 as a primary pharmacological target for vancomycin, instead of an inaccurate approximation using SVCs that poorly correlate to exposure”. Indeed, Berthaud et al. evaluated PK efficacy with AUC/MIC ratio. Vancomycin serum concentration target was a secondary endpoint in this study.

The authors should harmonize their statements on this point.

Line 173: “which correlates with improved clinical efficacy”

This type of statement should be discussed, especially when it refer to a case report or a small case series without control group.

The verb tense should be preterite.

Line 185-186: “The first one [20], compared the efficacy between patients treated with vancomycin CIV with (intervention n=16) and without (control n=27) an early Bayesian dose adjustment.”

These numbers (n=16 and n=27) seem to be wrong. They may refer to something else in berthaud et al article like the proportions of patients who reached vancomycin concentration target (see citation below) but if this is the case, this information should not be here and the numbers have been switched. “Twenty-seven patients (68%) from the Bayesian group versus 16 (38%) from the control group reached the H24 vancomycin concentration target of 20 to 40 mg/liter in the mITT population (P = 0.009)”

The authors should clarify this.

Line 190: “with a decrease in the frequency of exacerbations after the CIV therapy”

The statement should be rechecked and discussed, rephrased or removed. How could the frequency of exacerbations could decreased after the CIV therapy?

Line 191-193: “They observed clinical improvement in all six patients, with four of these also showing microbiological cure.”

This type of statement should be discussed, especially when it refer to a case report or a small case series without control group.

Line 220-221: “since this seems to facilitate the achievement and maintenance of therapeutic 220 SVCs avoiding high nephrotoxic doses as shown in this review.”

Are the authors really able to make this statement based on their review ?

Line 242-243: “Higher final total CIV dosing was used in the younger age groups (children < 2 years)”

Are the authors able to state this? Was the difference statistically significant?

Line 288-290: “A randomized controlled trial[20] compared clinical outcomes between Bayesian and control groups, both with CIV, and there were no differences between groups.”

The fact that this trial was not designed to evaluate this should be recalled here.

Line 350-352: “The systematic search strategy was conducted using a combination of search terms for the concepts: a) Vancomycin b) Pediatric OR Child OR Children OR Infant and c) Continuous infusion.”

The authors should detail the strategy that they used and the combinations of terms that brought them to the 359 records that they found. In the current state of the strategy description, it is not reproducible.

Reviewer 2 Report

The systematic review is well written and I do not have comments.

Author Response

We thank the Reviewer for their interest in our work.

Reviewer 3 Report

414 / 5000 It should be discussed that target trough concentration in intermittent infusion cannot be extrapolated to continuous infusion. Thus, for a target of AUC/MIC> 400-600 a trough concentration of 10-15 mg/L is accepted in mild-moderate infections and 15-20 mg/L in severe infections or those with difficult access. However, in continuous infusion a concentration of 20-25 mg/L is recommended for an expected AUC of 480-600.

Author Response

Comments and Suggestions for Authors:

414 / 5000 It should be discussed that target trough concentration in intermittent infusion cannot be extrapolated to continuous infusion. Thus, for a target of AUC/MIC> 400-600 a trough concentration of 10-15 mg/L is accepted in mild-moderate infections and 15-20 mg/L in severe infections or those with difficult access. However, in continuous infusion a concentration of 20-25 mg/L is recommended for an expected AUC of 480-600.

We thank the Reviewer for their interest in our work and the helpful comment that will greatly improve the manuscript. We have included in the discussion this statement:

“Trough SVCs in IIV cannot be extrapolated to CIV. Thus, for a target of AUC/MIC> 400-600 a SVCs of 10-15 mg/L is accepted in mild-moderate infections and 15-20 mg/L in severe infections or those with difficult access. However, in CIV a concentration of 20-25 mg/L is recommended for an expected AUC/MIC of 480-600 (assuming a MIC of 1 mg/L) [5].” (line 268-272)

Reviewer 4 Report

This systematic review triggers mixed feelings about the usefulness of the analysis. The Authors has not made a sufficient effort to justify why they included nearly all possible types of studies and how this translates into the quality of their reasoning about the efficacy and safety of CIV. Is it at all feasible to make valid assumptions about the efficacy and safety according to modern requirements for the evidence synthesis using retrospective studies? In the result, it seems that the value of the review is questionable as actually confirmed by ‘inconclusive’ conclusions. The justification for such review should have been made in the stage of prescreening.  And seemingly it has not been made.

The process study selection is not described in detail. It is not clear how papers included in the review were selected. It is not explained how the team participated in the procedure. Subchapter 2.1 should be included in the Materials and Methods and supplemented with information on the procedure and involvement of researchers.

A study selection flowchart should be also included in the Materials and Methods section.

Round 2

Reviewer 1 Report

I thank the authors for their answers to my comments.

Two points:

Line 82-84: The blind evaluation process should then be described.

Line 191-193 and 220-221: I understand the authors demonstrations, but case reports are not evidence. Not here or anywhere else. Confounding factors are not taken into account.

Reviewer 4 Report

The Authors responded to some degree to the comments from the first review. The reservations about the rationale of the study are still present to a considerable degree.  The addressed problem may be important in terms of medical practice, but it does not legitimate methodological failures and unsure integrity of aims. I believe the paper may be published to document for the scientific community the attempts of addressing the main topic.